# Co-Design of a Reusable Learning Object (RLO) to Address Caregiver Responsive Infant Feeding Behaviours (CRIB) to Prevent Childhood Obesity: A Mixed-Method Protocol

**DOI:** 10.3390/healthcare12010029

**Published:** 2023-12-22

**Authors:** Lucy Porter, Karen Matvienko-Sikar, Heather Wharrad, Helen Spiby, Aloysius Niroshan Siriwardena, Ciara Howitt, Katie Green, Sarah Redsell

**Affiliations:** 1School of Health Sciences, Queen’s Medical Centre, University of Nottingham, Nottingham NG7 2HA, UK; heather.wharrad@nottingham.ac.uk (H.W.); helen.spiby@nottingham.ac.uk (H.S.); ntykg29@nottingham.ac.uk (K.G.); 2School of Public Health, Western Gateway Building, University College Cork, T12 XF62 Cork, Ireland; karen.msikar@ucc.ie; 3School of Health and Social Care, University of Lincoln, Brayford Pool, Lincoln LN6 7TS, UK; nsiriwardena@lincoln.ac.uk

**Keywords:** responsive infant feeding, childhood obesity, reusable learning object, digital resource, intervention development, COM-B, co-design, protocol

## Abstract

Responsive infant feeding is a key strategy for childhood obesity prevention. Responsive feeding involves caregivers responding to infant hunger and satiety cues in a timely and developmentally appropriate manner. There is a dearth of evidence-based information and guidance for caregivers on how to responsively feed their infants. The aim of this research is to co-design a Reusable Learning Object (RLO) and guidance infographic to improve caregiver awareness, understanding and use of responsive infant feeding behaviours. The Capability, Opportunity, Motivation and Behaviour (COM-B) model of behaviour change and the Aim, Storyboarding, Populate specification, Implement media, Review and release prototype, and Evaluate (ASPIRE) approach for digital intervention co-design will be utilised. Four co-design workshops with caregivers of infants and healthcare professionals (HCPs) will determine priority RLO content. Content analysis will enable RLO development and process reporting. Formative and summative surveys will be conducted to evaluate the usability of the RLO, its impact on caregivers and its potential implementation into NHS care pathways. The output will be a RLO on responsive feeding for caregivers and an infographic for HCPs/support workers which will contribute to a future obesity prevention intervention. The findings will be disseminated to stakeholders and submitted for publication in a peer-reviewed journal.

## 1. Introduction

Globally, 39 million children under the age of 5 are overweight or obese [1]. There are substantial health inequalities in childhood obesity with higher obesity rates in children from families of lower socioeconomic status [2]. In the UK, obesity rates are twice as high for children living in the most deprived areas and are higher for black children [3]. During the first two years of life, infants develop rapidly and multiple inter-connected factors influence feeding behaviour, making this a crucial time period for preventing obesity [4].

The aetiology of childhood obesity is complex. Childhood obesity is linked with the infant feeding environment and in turn, what and how caregivers feed their infants is influenced by their socioeconomic status [5]. Overfeeding within an infant’s first year of life can lead to rapid weight gain, which is the largest risk factor for childhood obesity and overweight [6]. Formula feeding leads to a higher risk of rapid weight gain, regardless of socioeconomic status or extent of breastfeeding [7,8]. Lower maternal sensitivity to infant cues has been associated with rapid weight gain in 6–12-month olds [9]. Infants have been shown to have healthier weight gain trajectories when their caregivers are more responsive to their feeding cues [10]. Responsive parenting forms the basis of attachment between caregivers and infants and is essential for healthy socioemotional development and relationships [11]. Responsive feeding is the reciprocal relationship between infants giving clear hunger and satiety cues and their caregivers responding to these feeding cues promptly and in a developmentally appropriate way [12]. Such supportive early interactions with caregivers can enhance self-regulation, resulting in feeding autonomy and regulatory capacity in infants [13]. There are obesity prevention initiatives and interventions globally that focus on responsive feeding. The US INSIGHT study which delivered a responsive parenting intervention [14] showed reduced use of non-responsive feeding practices such as pressuring an infant to finish a bottle when compared to a control home safety intervention. Interventions delivered at an individual level, where responsive feeding is promoted and supported by healthcare professionals (HCPs), have shown larger improvements for both feeding and weight outcomes than interventions that do not focus on responsive feeding [15]. The World Health Organization (WHO) recommends [16] exclusive breastfeeding for the first six months followed by the introduction of nutritionally adequate and safe complementary (solid) foods. Breastfeeding is a preventative strategy for obesity but many families have negative experiences [17] and/or choose to use formula or combination feed (breast and formula milk). Formula milk companies provide recommendations about the volume of milk to offer an infant relative to their weight but this often vies with parents’ reading of infant cues [18]. Internationally, there are some sources of information for caregivers about responsive feeding. An example is the UNICEF information sheet on responsive feeding [19]; however, this largely focuses on responsive breastfeeding. The American Academy of Paediatrics also have a responsive feeding infographic [20]. However, existing resources do not appear to have been developed using a systematic approach addressing caregiver behaviours and the physical, social, cultural and environmental context [21], nor developed with caregivers.

The UK Childhood Obesity Plan aims to reduce childhood obesity by half by 2030 and focus on reducing health inequalities between obesity rates in the most deprived and least deprived areas of the UK [5]. The National Institute for Health and Care Excellence (NICE) guidance for postnatal care in England recommends that all caregivers, regardless of feeding mode, are given advice about responsive feeding [22]. However, at present, there is a lack of provision of responsive feeding guidance for caregivers, especially for formula and combination feeding in the UK National Health Service (NHS).

The UK Medical Research Council (MRC) Framework for the Development and Evaluation of Complex Interventions [23] highlights that intervention development should be underpinned by appropriate theory. The Behaviour Change Wheel (BCW) [21,24] provides such a framework for developing and evaluating behaviour change interventions. Central to the BCW, the Capability, Opportunity, Motivation and Behaviour (COM-B) model acknowledges that capability (physical and psychological), opportunity (physical and social) and motivation (reflective and automictic) are key factors to be addressed to create effective behaviour change interventions [21]. The use of the COM-B framework allows behaviours to be identified and targeted by an intervention.

Barriers and enablers to responsive feeding for caregivers were identified in a systematic review of responsive feeding behaviours [18]. Barriers and enablers within five COM-B domains were identified: psychological capacity, physical and social opportunity, reflective and automatic motivation. For example, barriers to responsive feeding including not recognising developmental signs and feeding cues, having feeding goals, feeling stigmatised by HCP attitudes for formula feeding and also written instructions on formula opposing responsive feeding practices can discourage responsive feeding. Enablers such as caregivers being able to recognise infant feeding cues and being given advice and support from HCPs, family and peers were identified as facilitating responsive feeding. This systematic review identified the behaviours to target within an intervention to improve responsive infant feeding. It also highlighted the need for responsive feeding interventions for caregivers to be developed in tandem with guidance for HCPs. Such HCPs include midwives, health visitors and support workers such as family mentors who work with families to provide advice and support including around infant feeding.

Having established the evidence base for responsive feeding intervention development, the MRC Framework [23] further emphasises engaging and working with stakeholders at each stage of designing and conducting intervention research to maximise the likelihood that interventions are fit for purpose and have positive impacts on health outcomes. Therefore, the co-creation of interventions with caregivers and HCPs such as health visitors, infant feeding leads and support workers who work with caregivers and offer infant feeding advice and support is vital to support responsive feeding. The framework also highlights considering the implementation of the intervention during its development to increase the likelihood that the intervention can be effectively transferred into real-world settings.

Caregivers seeking health information for their children have reported using the internet as their main source of information and reported high levels of confidence [25], and individuals seeking health information will most often use mobile devices [26]. However, a review of websites on infant health promotion including infant feeding found that the quality, accessibility and readability of the information was poor or adequate [27], highlighting the need for the development of interventions for caregivers and HCPs to be evidence-based, appropriate and useable.

The Public, Patient, Involvement and Engagement (PPIE) group who informed the development of this funding application suggested that any intervention around responsive feeding should utilise digital technology. Therefore, this research aims to develop an app to deliver a behaviour change intervention to caregivers. The chosen approach is a Reusable Learning Object (RLO) which consists of an interactive format with audio, text, animations and videos to promote health behaviour change. To this end, this research will co-create a RLO with caregivers to promote responsive feeding.

The aim of this study is to develop a RLO for caregivers of infants under one year to increase awareness, understanding and use of responsive infant feeding behaviours (CRIB). Improving responsive feeding will contribute to the prevention of obesity at an individual and population level. In addition, this study aims to develop infographic guidance for HCPs and support workers to use with caregivers alongside the RLO.

Objectives:To co-design a RLO to increase caregiver awareness, understanding and use of responsive infant feeding behaviours.To create an infographic for HCPs and support workers to improve their knowledge and ability to support caregivers with responsive feeding.To conduct an initial evaluation of the acceptability of the RLO for caregivers and the guidance infographic for HCPs and support workers.To determine the care pathways within the NHS, social care enterprises and the local authority where the RLO will be offered to caregivers with appropriate support.

## 2. Materials and Methods

### 2.1. Design

This is a mixed-method study involving qualitative co-design workshops with caregivers and HCPs/support workers and a combination of quantitative and qualitative approaches to inform RLO development and evaluation. A mixed-method design enables us to gather comprehensive information from both caregivers and HCPs beyond that which would be possible from a single quantitative or qualitative approach. In addition, the use of qualitative, interactive approaches enables more in-depth and rich data which are complemented by the broader reach of data captured via the survey approaches used. This study does not involve quantitative hypothesis testing and instead involves an exploratory survey approach where sample size calculations have not been conducted. This study will instead aim to recruit a broad representative sample of participants to address the research aim.

The BCW framework [21] and the Behaviour Change Techniques Taxonomy (BCTT) [28] will be utilised to co-create a behaviour change RLO with caregivers, HCPs and support workers. The Am, **S**toryboarding, **P**opulate specification, **I**mplement media, **R**eview and release prototype, **E**valuate (ASPIRE) approach [29] for designing digital education resources will be used, including stakeholder involvement in co-designing the RLO and infographic for HCPs and support workers. As part of the final evaluation step, preliminary summative and formative evaluation of the usability, acceptability and implementation of the RLO and infographic will be undertaken.

Due to the lack of guidelines for the reporting of mixed-method research, the Consolidated Criteria for Reporting Qualitative Research (COREQ) checklist [30] will be followed in conducting and reporting this research.

### 2.2. Data Collection

We will follow the COM-B method to develop the RLO and infographic [24]. Following the COM-B framework for intervention development consists of three phases; we have added an additional phase to enable us to evaluate the RLO and infographic resources developed. The phases (see Figure 1) to be followed in this project are as follows:Understanding the behaviour;Identifying intervention functions and BCTs;Determining BCTs and content options;Formative and summative evaluation of the RLO and infographic.

Phases 1, 2 and 3 will be conducted over four workshops with caregivers and HCPs/support workers. This study will use an iterative approach whereby data collection and analysis will be conducted in tandem between workshops and between phases. Field notes taken by the facilitators, flipchart data, sticky notes, storyboard data in text, and image data gathered within the four workshops will be collated and used throughout the study to inform the data collection within the next phase. Phase 4 is the final phase and will consist of a formative and summative evaluation of the RLO and infographic created as a result of the workshops.

### 2.3. Data Collection

This project will take place in the city of Nottingham, UK. Nottingham City has high levels of social deprivation and minority ethnic groups. It ranks 11th most deprived out of the 317 districts in England [31]. According to the 2021 census, 65.9% of Nottingham residents identified their ethnic group within the White category, 14.9% identified within the Asian category, 10.0% within the Black category and 5.9% within the Mixed or Multiple category, with the remaining 3.3% identifying within the Other group [32].

Recruitment to workshops (phases 1 and 3)

We will recruit participants from two groups to the workshops. These participants will include caregivers (parents, guardians and carers) of healthy infants ≤ 1 year of age, and HCPs working with caregivers (health visitors and infant feeding leads) and support workers (family mentors from Small Steps Big Changes (SSBC)). SSBC is a programme of work under the national lottery-funded Better Start Scheme (https://www.smallstepsbigchanges.org.uk/) that provides support to caregivers in the most socioeconomically deprived areas of Nottingham. To maximise engagement and to minimise attrition, we will offer potential participants shopping vouchers to cover their childcare costs, taxis to and from the venue and lunch. The workshops will take place in a child-friendly environment with space for caregivers to take time out and with changing facilities. Participants will be asked to participate in all four workshops, which will take place in phase 1 and phase 3.

*Caregivers.* The sample will consist of caregivers (parents, guardians and carers) ≥ 18 years of age who are the main carer of an infant < 1 year of age at the time of recruitment. Caregivers will be recruited via SSBC. NICE guidance refers to the need to include people with all perspectives and skills for co-design rather than a specific number [33]. These are face to face workshops and it is vital that all voices are heard; we will recruit up to ten caregivers to ensure all participants can contribute meaningfully to the discussions. We will aim for maximum variation in gender, age, socioeconomic background and ethnicity and will use a sampling frame to ensure representation.

Family mentors from SSBC who wish to support recruitment will display a study poster and offer the participant information sheet to potential stakeholders. Family mentors will pass on the contact details of participants to the RA.

The following caregivers are excluded from the study:<18 years of age;Infant age > 1 year;Has an infant who is not fed orally (nasogastric or parental nutrition);Has an infant with a condition(s) likely to impact their ability to feed, since this impacts on how caregivers feed their infants and the supports needed;Caregivers who do not have access to digital technology.

*HCPs and support workers.* The RLO will be developed by caregivers for other caregivers. We are also keen to include the perspectives of HCPs (health visitors, infant feeding leads and family support workers) in the workshops to contribute to our understanding of what might be possible in relation to the provision of infant feeding advice in the NHS and social care. HCPs will be recruited from local NHS trusts, SSBC and social care enterprises. We will aim for at least one representative health visitor, infant feeding lead and family support worker to attend our workshops.

The following HCPs (and support workers) are excluded from this study.

HCPs/support workers who are not working with caregivers of infants <1 year of age.

#### 2.3.1. Phase 1—Understanding the Behaviour

This phase involves identifying the who, what, when, where and how often of responsive feeding behaviour. We will undertake this work over two workshops.

*Workshop 1*. Following introductions and ground rule setting, especially around a non-judgemental approach to infant feeding decisions, we will raise the idea that everybody’s views are equally important to ensure that each group’s (caregivers and HCP/support workers) views are equally heard. We will familiarise participants with the concept of responsive feeding by explaining what it means and what it involves. We will help them to consider the behaviours that are involved through time to reflect and group discussion. Brainstorming will be encouraged to generate a long list of all of the possible relevant behaviours involved in the process of responsive feeding. Caregivers, HCPs and support workers will feedback to the full group visually (using flipcharts) and verbally. All participants will be asked to rank these behaviours by considering a) how much of an impact change in each behaviour will have on responsive feeding, and b) the likelihood of change in behaviour, using post-it notes/coloured pens/sticky dots. At the end of the session, we will produce a list of caregiver-prioritised target behaviours.

The qualitative data from participant discussions during the four workshops will be audio-recorded by a dictaphone, saved in MP4 format and transcribed verbatim by the RA. Transcriptions from audio recordings from the four workshops will be exported to NVIVO 14 for analysis. Qualitative content from the workshops will be anonymised during transcription. Content analysis using guidance from Elo and Kyngäs [34] of the four workshops will be conducted by the Researcher (LP) and data from two workshops will be checked for accuracy by the PI. Content analysis will include four steps: familiarisation with the data by reading and re-reading the transcripts; NVivo 14 software will be used for open-line coding; categorisation and development of line code; grouping of categories and descriptions created with categories and sub-categories. The content analysis of each workshop will be used to inform the next phase of the RLO development process and to write up the full process from phases 1 to 3.

Following workshop 1, the research team and PPIE lead will map the priority behaviours to the COM-B model [21]. This involves applying each of the participants’ target behaviours to one element of the COM-B framework: capability (physical and psychological), opportunity (physical and social) and motivation (reflective and automatic). For example, if ‘knowing how to read babies cues’ is identified as an important behaviour, we would map this to psychological capability. We will also add the barriers and enablers identified and map them to one element of the COM-B model within our systematic review [18].

*Workshop 2.* We will identify the COM-B component areas for change supported by the priority target behaviours identified in workshop 1. We will summarise the behaviours identified in workshop 1 in the COM-B context for presentation and discussion in workshop 2. Caregivers and HCPs/support workers will be separated into two groups and asked to specify the behaviours involved. For each of the COM-B components, caregivers will be asked to consider who might carry out this behaviour (caregiver or HCP) and what might need to happen for caregivers to change their behaviour in this regard (greater attention to infant cues, when (mealtimes), where (in the family home) and how often (at every mealtime). HCPs and family mentors will be asked to consider these in terms of their advice giving, who (family mentor), what (specific advice), when (after birth), where (in clinic) and how often (weekly).

Discussions will take place around the barriers to and enablers of each of the previously identified behaviours. The group will be prompted by asking what might make each of the behaviours easier to carry out, for example, being able to recognise hunger cues, and what might make each of the behaviours harder to carry out/might compete with being able to responsively feed, for example, being distracted by other people in the room [18]. Caregivers and HCPs/support workers will feedback to the full group on the barriers and enablers identified, discussing any solutions and contextual factors. We will identify which solutions could be incorporated into the RLO and the wider contextual behaviours. At the end of the session, we will produce a contextual specification for each of the identified specific behaviours, and the potential enablers and the solutions to the barriers. The behaviours and barrier solutions will be separated by appropriateness for the caregiver RLO or the HCP infographic.

#### 2.3.2. Phase 2—Identifying Intervention Functions and BCTs

##### Method and Analysis

Following phase 1, phase 2 will consist of mapping the COM-B components identified in phase 1 to intervention functions. Within the BCW, intervention functions include nine broad categories such as education and environmental restructuring, which are evidence-based principles of how an intervention can change behaviour. The COM-B components have been linked with specific intervention functions that most are likely to result in behaviour change. This detailed mapping will be conducted by the research team and the PPIE lead, giving an output of intervention functions to target within the RLO.

Intervention functions will also be mapped to individual behaviour change techniques (BCTs) [28] which are “an active component of an intervention designed to change behaviour” [24] such as goal-setting or self-monitoring. We will identify the most frequently used BCTs for each intervention function to present to the caregivers and HCPs for discussion in workshop 3.

The output of phase 2 will be a list of potential content and proposed BCTs for the RLO and the infographic to promote discussion in workshop 3.

#### 2.3.3. Phase 3—Determining Behaviour Change Techniques and Content

Phase 3 will be carried out via workshops 3 and 4.

##### Method and Analysis

*Workshop 3.* We will decide on the most appropriate and feasible intervention components including BCTs and content. The potential content and BCTs developed in phase 2 will be presented to the caregivers and HCPS/support workers by the workshop facilitators. The identified intervention functions will be checked against each of the six APEASE criteria [21]: affordability, practicability, effectiveness and cost-effectiveness, acceptability, side effects/safety and equity. The whole group will consider each individual intervention function against each of the six APEASE criteria to help identify the most appropriate intervention functions to use in the RLO.

The facilitators will present all BCTs that could be considered for each intervention function, identified within phase 2. The number of BCTs will be reduced by using each of the APEASE criteria to find the most appropriate and feasible BCTs to use in the RLO. We will reach a consensus on which BCTs to include in the RLO by ranking each of the ideas by preference to use within the RLO.

The output of the workshop will be a consensus of what should be included in the final RLO and how this should be delivered.

*Workshop 4.* The participants will include caregivers, HCPs and technologists from the University of Nottingham’s health e-learning and media team (HELM) and will use the ranked topic ideas and BCTs from workshop 3 to work on how these can be represented pedagogically and visually in the RLO. The workshop will explore with the participants any analogies, anecdotes, images, videos and personal stories that will engage learners and facilitate their understanding of the ideas and guidance. A0 laminated storyboard templates will be provided for participants to express their ideas in text or drawings. Facilitators will direct the participants using a crib sheet outlining the Theoretical Domains Framework (TDF) with COM-B in order to identify the essential components or ‘active ingredients’, i.e., the visual and graphic elements that represent the knowledge, skills and motivational aspects incorporated into the RLO to bring about the desired behaviour change (responsive feeding)—for example, some of the components that the RLO might address are shown in Table 1.

The output of this workshop will be the creation of a storyboard of caregiver responsive feeding behaviours. In addition, this workshop will concentrate on how to translate non-caregiver behaviours into practice.

Following workshop 4, a prototype of the RLO for caregivers will be developed by the HELM team following the ASPIRE process, including populating a detailed specification from the storyboards which will be peer reviewed by members of the advisory committee. The ‘active ingredients’ in the form of various media assets such as videos and interactivities will be produced guided by the storyboard ideas and inserted into the RLO template along with audio and text components. The design of the RLO will be based on the output from the four workshops integrating the required behaviour changes for responsive feeding. The HELM team will also develop guidance for HCPs/support workers in the form of an infographic, integrating images and key messages from the RLO as appropriate, to accompany the RLO for caregivers. A radar diagram will map the segments of the RLO that relate to the specific domains of the theoretical domains framework as a research output of the study.

#### 2.3.4. Phase 4—Formative and Summative Evaluation

The final phase will be a formative and summative evaluation of the RLO and infographic following the ASPIRE [29] methodology.

##### Recruitment for Evaluation

We will conduct a preliminary evaluation of the functionality, acceptability and usability of the RLO. Caregivers, HCPs and support workers (who were not part of the RLO development) will be recruited for either a formative survey or one of two summative surveys. Potential participants will be emailed a link to the RLO, patient information sheet (PIS), informed consent form (ICF) and respective survey. Participants will be given up to two weeks to access and complete the surveys and will be able to work through the RLO as many times as they require during this two-week period.

Survey 1—Up to 20 caregivers, HCPs and support workers will be asked to review the prototype RLO for content and ease of use.

Survey 2—Up to 30 caregivers will be recruited via Small Steps Big Changes (https://www.smallstepsbigchanges.org.uk/) and public contributors.

Survey 3—Up to 30 HCPs and support workers will be recruited via contacts at the Institute of Health Visiting and the Royal College of Midwives and gatekeepers at local NHS trusts who will be asked to send the survey via email to HCPs and support workers.

##### Method

In the first step of the formative evaluation, public contributors, content experts and members of the healthy weight/healthy nutrition group at the Institute of Health Visiting will be asked to evaluate the proposed content to ensure that it is accurate. The public contributors include caregivers from a diverse range of backgrounds who have made different feeding choices (breast and/or formula). The RLO specification will be amended following this evaluation, and the content will be uploaded to the media platform. The prototype will be evaluated by public contributors to ensure that it is fit for purpose by looking at the fit between the content and media.

Next, an initial analysis will be conducted after completing the RLO through three online, anonymous surveys evaluating the acceptability and usability of the RLO. Participants will view and engage with the RLO, which typically takes 15–20 min, although they can review sections as many times as they wish. The anticipated time for completion of each survey is no more than 20 min.

*Survey 1* (formative) responses will explore the usability of the RLO content and functionality and will be completed by caregivers, HCPs and support workers. The questions have been adapted from a survey tool by Johnson et al. exploring the usability of an e-learning resource to improve the knowledge and confidence of teachers working with children born pre-term [35]. Survey 1 (see Appendix A Table A1 survey A1) will consist of 9 closed questions with responses of agree, neutral or disagree to questions such as “the CRIB-RLO resource held my interest”.*Survey 2* (summative) (see Appendix A Table A2 survey A2) will be completed by caregivers and will ascertain the impact of the RLO on caregivers in terms of changing behaviours related to responsive feeding. There will be 7 closed questions on a 3-point scale from agree, neutral and disagree, including questions such as “The CRIB-RLO resource has improved my confidence in knowing how to feed my baby” and “The CRIB-RLO resource has increased my motivation to actively look for and respond to my babies cues during feeding”.*Survey 3* (summative) (see Appendix A Table A3 survey A3) will be completed by HCPs working with caregivers of infants and will explore the implementation of the RLO and infographic in practice within the NHS and other organisations. The questions have been developed based on a the Theoretical Framework of Acceptability (TFA) developed by Sekhon et al. [36]. The TFA can be used to evaluate the prospective and retrospective acceptability of healthcare interventions by both intervention deliverers and intervention participants. Preliminary reliability testing on the Sekhon et al. TFA adapted for examining the implementation of a telephone-assisted coaching framework revealed three main factors. These are affective attitude and effectiveness (Cronbach’s alpha 0.9, coherence and understanding (0.77), perceived burden (0.85), and suggesting an appropriate level of internal consistency for each construct [37].

Survey 3 will include a mixture of eight open and two closed questions on the implementation of the RLO in practice. Open questions include those such as “Can you describe any additional effort you might need to employ to implement the CRIB-RLO/Infographic into practice?” and closed questions such as “how prepared are you to deliver the CRIB RLO to caregivers?” with a 5-point Likert scale ranging from not at all prepared to highly prepared.

For a long-term evaluation beyond the term of the project, it is standard practice for HELM to monitor RLO feedback from users through an optional short feedback survey at the end of the RLO. This will track ongoing usage and utility of the RLO. Since this survey is optional, Google analytics are also used to track global usage and reach.

##### Analysis

Survey responses will be downloaded to an Excel/Sav file.

Survey 1 responses evaluating RLO content, functionality and usability will be analysed with descriptive statistics in the form of frequencies and percentages. The data will be categorised to highlight areas of function or content that require amending.Survey 2 responses examining the impact and usefulness of the RLO on caregivers will be analysed with frequency tables and descriptive statistics. Data from the closed questions will be analysed to examine the impact of the RLO on changing behaviours related to responsive feeding.Survey 3 responses from HCPs exploring the implementation of the RLO and infographic will be a mixture of quantitative and qualitative data. The quantitative data from the two Likert scale questions will be analysed with frequency tables and descriptive statistics. The qualitative data gathered from the open questions will be analysed using content analysis [34].

### 2.4. Workshop Facilitator Roles and Experience

LP is working as a research assistant (RA) and has an MSc in Health Psychology. She has completed training in qualitative research methods, focus groups and behaviour change interventions. She has previously worked as a RA on behaviour change intervention trials and has no personal experience of infant feeding.

SAR is a health visitor and a health psychologist. Sarah has led the systematic review looking at barriers to and enablers of responsive infant feeding and is interested in a population-level approach. Sarah set up the PPIE group associated with this project and believes in a balanced approach to the provision of infant feeding information. She has been the mother of a pre-term infant who she breastfed.

KMS is a health psychologist with expertise in infant feeding and childhood obesity. She has developed an infant feeding intervention for use in primary care: the Choosing Healthy Eating for Infant Health (CHErIsH) intervention. She has also developed core outcome sets of infant feeding outcomes for childhood obesity prevention interventions and of outcomes in children up to 5 years in childhood obesity prevention interventions.

HW is a specialist in health informatics and digital learning. She developed a co-creation method for producing digital resources and has contributed to many projects using this approach. She breastfed her own children and now has grandchildren who were breast and formula fed.

CH is the PPIE lead and a young parent of two. She has both current and recent experience of infant feeding by both breast and bottle. She has two children under the age of 3—one exclusively breastfed and one fed with a combination of both breastmilk and formula.

KG is a child nursing student who has been working on an internship at the university. Katie has a keen interest in health visiting and supporting families in their journey. She is the mother of a child who was exclusively breastfed for the first six months and then breastfed in tandem with weaning until nine months, when their child was moved to formula.

### 2.5. Data Storage

All data will be treated as confidential and stored according to the GDPR 2018, Data Protection Act 2018. Working data will be stored on the University of Nottingham’s provided storage Microsoft Teams to enable data to be shared within the research team. Research data will be stored on the University of Nottingham’s OneDrive.

## 3. Results

### Dissemination

The findings including a description of the final RLO will be written up and published in a peer-reviewed journal. The findings will be presented at academic conferences in the areas of infant feeding, health psychology and digital health, and to stakeholders such SSBC. Virtual knowledge exchange and dissemination events will be held virtually with HCPS, caregivers and charitable NHS representatives. The RLO for caregivers will be circulated via public contributors, social media and parenting organisations such as the NCT, who will be asked to provide a link to the RLO via their webpages. Professional bodies will be made aware of the RLO in order to direct HCPs/support workers to pass the RLO on to caregivers. The accompanying infographic for HCPs and support workers will be disseminated via professional networks including the Royal College of Midwives, the British Nutrition Foundation and the Institute of Health Visiting. Higher education institutions that train midwives and health visitors will be sent a link to the infographic. The RLO will be uploaded onto the HELMOpen repository (www.nottingham.ac.uk/helmopen, accessed on 1 October 2023) which provides free global access to RLOs and has registered more than 5 million users. The usage is tracked via Google Analytics, and optional feedback forms are routinely collected as measures of impact.

## 4. Discussion

Whilst population-based approaches are needed to address the multiple and complex causes of childhood obesity, individual prevention efforts that support parents with infant feeding make a useful contribution. There is strong evidence that individual obesity prevention interventions delivered during early life that promote and support responsive feeding result in healthier weight trajectories for infants [15]. However, there is no universal information and support focused on responsive feeding; thus, there is a need to improve information for caregivers who feed infants and the HCPs who work with them. This study will co-produce a RLO for caregivers of infants to increase awareness, understanding and use of responsive infant feeding behaviours. In addition, we will developed infographic guidance for HCPs and support workers to use with caregivers alongside the RLO.

This research will been conducted in a comprehensive, systematic and evidence-based way by applying the MRC framework for complex interventions [23], the COM-B model and the ASPIRE approach [29]. This will help to ensure that the developed RLO and infographic responsive feeding outputs are robust and acceptable to both caregivers of infants and the HCPs working with them.

The co-design and initial evaluation of the RLO and infographic will allow for future testing of the intervention effectiveness on both responsive feeding behaviours and child weight outcomes within a randomised control trial. Providing improved support for responsive feeding for caregivers and HCPs who work with them has the potential to reduce non-responsive feeding practices such as overfeeding, which can lead to obesity.

## Figures and Tables

**Figure 1 healthcare-12-00029-f001:**
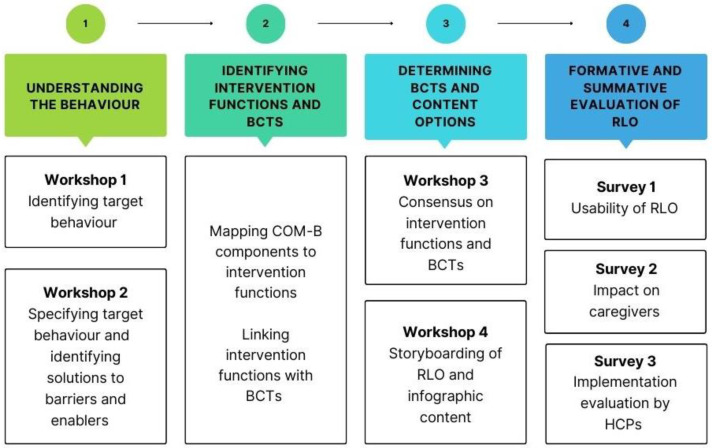
Outline of study phases and components.

**Table 1 healthcare-12-00029-t001:** Potential RLO components, COM-B elements and TDF domains.

Topic Idea	COM-B Component	TDF Domain
How do I work with others to help me carry out responsive feeding?	Opportunity	Social influences
What can I change around me to improve my ability to responsive feed?	Capability	Behavioural regulation; Environment
What are the cues for hunger/fullness from my baby?	Capability	Knowledge/Skills
Why should I carry out responsive feeding?	Motivation	Goals/Intentions

## Data Availability

Data created and analysed in this study will inform the RLO development. Data sharing is not applicable to this article.

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
