# Peer review of "Co-Design of a Reusable Learning Object (RLO) to Address Caregiver Responsive Infant Feeding Behaviours (CRIB) to Prevent Childhood Obesity: A Mixed-Method Protocol"

_healthcare, 2023, doi:10.3390/healthcare12010029_

Round 1

Reviewer 1 Report

Comments and Suggestions for Authors

I am very much looking forward to the product of this study. The commitment to including the community in research and to obesity prevention is clear. I would encourage the researchers to not limit their understanding of responsive feeding practices to obesity prevention, but to also put it in a wider context of relationship building, bonding, communication development and prevention of food aversion/ fussiness.

This is a very detailed description of the method that seems slightly at odds with the idea of co-production. Given the involvement of the caregivers and HCPs is to contribute to the content of an app, I think this should be made clearer. The format of the information i.e. an app and an infographic, is already determined. I would like to see some mention of alternative potential or unintended outcomes. 

Line 73-74 - Regarding the systematic approach. Several of the available feeding cue charts include early-, mid- and late-cues. Please can you be more specific about what you are expecting of a systematic approach and what you mean by the wider feeding context. Are you referring to cultural practices around infant feeding?

Given the international audience of this journal, I'd like to see a description of the demographic your participants are coming from. How likely is it that you can recruit 10 participants to attend all four meetings. Will you set a quorum? Will you offer incentives? 

In phase 4 you say there are three summative surveys, but I think there are two summative and one formative.

Regarding the 20 min time frame, does this include using the RLO? How much time do you anticipate people will spend using the app, before they can give constructive feedback?

You haven't said who/ what HELM is. Can you design the app to capture data about usage, which could inform future iterations?

Survey A1 seems appropriate if the respondents agree with the statements. However, as this is formative, it does little to provide feedback if the respondents disagree. There is no option for the respondents to offer suggestions to say why something doesn't work for them, or what could be improved and how.

Will there be any training or additional information for the HCPs - is that the purpose of the infographic? this is unclear.

How much time will respondents have access to the app for before they complete surveys 2 and 3? Survey A2 is very open to social responding bias. A more sophisticated or nuanced evaluation would be preferable. 

There are some grammar errors such as occasional extra words or a full stop in an unexpected place.

There is some repetition in the reference list e.g. Michie 2011. There are also capitalisation errors and doi is inconsistently formatted.

Comments on the Quality of English Language

This is fine.

Reviewer 2 Report

Comments and Suggestions for Authors

While the protocol of Porter and colleagues is an interesting read, there are several inconsistencies in the paper that dampen enthusiasm of readers. More clarity is needed in the following:

1. The title does not seem to match what the study is about. The objectives in the abstract and introduction also do not match. Its not clear whether an app will be developed or RLO as well as the mention of co-production.

2. I agree that mixed methods is an under utilized design. Hence, the authors are requested to justify why this is the most suitable design to meet the objectives.

3. Since there is a quantitative part the authors need to perform sample size calculation, otherwise they need to explicitly explain why this is not necessary.

4. The inclusion and exclusion criteria needs to be expanded.

5. The survey tools to be used need to be elaborated aside from just citing sources. Are these developed and validated tools? If yes what were the psychometric properties tested (reliability, validity, responsiveness, etc...)?

Comments on the Quality of English Language

Several confusing statements need editing for clarity.

Round 2

Reviewer 1 Report

Comments and Suggestions for Authors

Detailed responses to my comments were helpful and your amendments have improved the clarity of the manuscript. I wish you well with the project and future funding.

Reviewer 2 Report

Comments and Suggestions for Authors

I am satisfied with the authors' rebuttal.